# Effect of Vitamin D in Long COVID Patients

**DOI:** 10.3390/ijerph20227058

**Published:** 2023-11-13

**Authors:** Ramsen Ghasan Hikmet, Christian Wejse, Jane Agergaard

**Affiliations:** 1Department of Infectious Diseases, Aarhus University Hospital, 8200 Aarhus, Denmarkjaneager@rm.dk (J.A.); 2Center for Global Health, Aarhus University (GloHAU), 8000 Aarhus, Denmark; 3Department of Clinical Medicine, Aarhus University, 8000 Aarhus, Denmark

**Keywords:** COVID-19, SARS-CoV-2, long COVID, vitamin D

## Abstract

Vitamin D deficiency has been studied in the context of acute coronavirus disease 2019 (COVID-19), revealing associations with increased severity and mortality. Yet, the influence of vitamin D on long COVID symptoms remains unknown. The purpose of this study is to examine the effect of vitamin D on long COVID symptoms. Over the study period, 50,432 individuals within the catchment area of the outpatient COVID-19 clinic tested positive for SARS-CoV-2 via PCR, with 547 patients subsequently referred to a specialized Post-COVID Clinic, and 447 of them enrolled in the study. Patient-reported symptoms and paraclinical measures including vitamin D were evaluated in 442 patients. The majority of participants were female (72%, *n* = 320/442). The consumption of alcohol and number of current smokers were low. Low vitamin D was observed in 26% (*n* = 115/442) of the patients, most commonly in male participants (odds ratio (OR) = 1.77, 95% confidence interval (CI) (1.12, 2.79), *p* = 0.014). Additionally, low vitamin D was correlated with a younger mean age of 41 years (standard deviation (SD) = 12) as opposed to 48 years (SD = 13) in patients with normal vitamin D levels (OR = 0.96, 95% CI (0.94, 0.97), *p* < 0.001). While our study population indicated a potentially higher prevalence of vitamin D insufficiency in this population compared to the general population, no significant differences in prevalence of symptom or symptom severity scores were observed between the low and normal vitamin D groups. In patients in a Post-COVID Clinic, we found no association between vitamin D levels and long COVID symptoms.

## 1. Introduction

The coronavirus disease 2019 (COVID-19) pandemic, caused by severe acute respiratory syndrome coronavirus 2 (SARS-CoV-2), swiftly evolved into a global health crisis, straining public health systems in multiple countries. With the introduction of vaccines, antivirals, and monoclonal antibodies [1], and the emergence of new SARS-CoV-2 variants, COVID-19 no longer poses a global health emergency [2]. While acute COVID-19 mortality and severity have significantly decreased, unresolved issues persist regarding the long-term sequelae of post-COVID-19 infection [3,4,5].

Various reports have revealed that a substantial proportion of individuals recovering from COVID-19 suffer from numerous health issues commonly referred to as long COVID. The symptoms of long COVID, often including fatigue, concentration difficulties, altered sense of taste and/or smell, memory problems, and shortness of breath [6,7].

The precise mechanisms by which SARS-CoV-2 infection leads to the development of long COVID in some patients remain elusive. Several hypotheses have been proposed, including persisting reservoirs of either components of, or active, SARS-CoV-2 in tissues [8]; SARS-CoV-2 binding to angiotensin-converting enzyme 2 (ACE-2) expressed by endothelial cells, leading to endothelial swelling and dysfunction, and impaired oxygenation in blood and tissues [9,10]; and autoimmunity triggered by molecular mimicry, where cross-reactive antibodies, due to similarities between SARS-CoV-2 antigens and human proteins, cause sustained damage [10,11].

Evidence suggests that myopathy induced by SARS-CoV-2 may contribute to the fatigue and muscle weakness observed in long COVID patients [12,13,14]. Additionally, disruptions in neurological signaling and alterations in the microbiota have also been observed in long COVID [10]. Given the extensive impact of SARS-CoV-2 on various systems of the human body, it is reasonable that the symptoms of long COVID are diverse.

Vitamin D plays multiple immunomodulatory roles. Studies have demonstrated that the active form of vitamin D3 (1,25-dihydroxycholecalciferol) can regulate both the innate and adaptive immune system through various pathways [15]. Vitamin D has also been shown to have a vasoprotective effect on the endothelium by activating endothelial nitric oxide synthesis (eNOS) and thereby regulating the eNOS synthesis [16]. Due to the widespread prevalence of vitamin D insufficiency or deficiency [17] and its immunomodulatory role, vitamin D has garnered interest for its potential role in combating infections, particularly respiratory viral infections. It has been shown that vitamin D is protective against acute respiratory tract infections, presumably through the induction of the antimicrobial cathelicidin peptides [18].

Recent studies have indicated that vitamin D supplementation may play a role in the prevention and treatment of COVID-19, as vitamin D deficiency has been linked with an increased risk of acquiring COVID-19 and an increased risk of severe COVID-19 and mortality [19,20,21,22,23,24,25]. Currently, no medicine has been proven effective for treatment of long COVID.

Given the intersection between vitamin D’s protective properties on the immune system, endothelial tissues, and protection against respiratory viral infections, and the destructive properties of SARS-CoV-2, it is logical to investigate the relationship between long COVID and vitamin D. Furthermore, vitamin D supplementation is a simple and cost-effective intervention. The aim of this study is to shed light on the possible association of low vitamin D levels and the manifestation of long COVID symptoms.

## 2. Materials and Methods

### 2.1. Setting and Participants

The study included patients who visited a specialized outpatient Post-COVID Clinic run by the Department of Infectious Diseases, Aarhus University Hospital, Denmark, from 1 February to 1 November 2021. The clinic serves a catchment area of one million people, among whom 50,432 individuals tested positive for SARS-CoV-2 using PCR from the beginning of the COVID-19 pandemic until 8 August 2021 (three months before the end of study period). During this timeframe, 547 out of the 50,432 individuals were assessed at the Post-COVID Clinic and diagnosed with long COVID (DB948A) [26].

Patients were referred to the Post-COVID Clinic by their general practitioner (GP), in line with guidelines from the National Board of Health, which required referred patients to have complex and prolonged symptoms for more than 12 weeks [27]. The patient cohort has previously been described by Agergaard et al., 2022 [26].

### 2.2. Data Collection

During the initial clinic visit, data on demography, medical history, and results from paraclinical investigations were recorded in the electronic patient journals (EPJ), using an interview guide. This data was subsequently entered into a secured REDCap database, along with patient-reported symptoms and standardized health scores. Immigrants encompassed both first and second generations of immigrants and were defined based on country of origin and language information collected using the interview guide. Non-western European ethnic origin included individuals from Iraq, Iran, Palestine, Lebanon, Turkey, Hungary, Russia, Croatia, Poland, Thailand, Sri Lanka, Korea, Peru, India, Somalia, Tanzania, and nonspecific African origin [28]. Insufficient vitamin D was defined as 25-OH D3 between 25 nmol/L and 50 nmol/L, and vitamin D deficiency was defined as 25-OH D3 < 25 nmol/L, with low vitamin D referring to 25-OH D3 values ≤ 50 nmol/L.

Various symptoms, typically considered organ-specific, were used to evaluate organ-specific severity, as described by Agergaard et al. [26]. These include the central nervous system (CNS) score (0–20), flavor score (0–8), cardiopulmonary (CP) score (0–20), gastrointestinal (GI) score (0–20), and musculoskeletal score (MS) (0–20).

Standardized, previously validated scores were used to assess shortness of breath (Medical Research Council (MRC) Dyspnea Scale), cognitive function (Orientation–Memory–Concentration (OMC)), functional disability (Post-COVID Functional Status Scale (PCFS)), fatigue (Fatigue Assessment Scale (FAS)), and mental health (SCL-13). Health-related quality of life was assessed using the EQ-VAS and EQ-5D-5L indices.

The seasons were divided into a summer season defined by the months April, May, June, July, August, and September, and a winter season defined by January, February, March, October, November, and December.

### 2.3. Data Analyses

The impact of low vitamin D values was assessed for each symptom, compound organ-specific scores, total scores, and standardized health scores. Symptom and health scores were described in our previous study [26]. In binary analyses, individual symptoms were categorized as “not present” if they were registered “none”, “a little”, or “unchanged compared to before the acute phase of COVID-19”.

Previously, a Post-COVID Questionnaire Symptom (PCQ) score, which is the sum of all 31 symptom scores, was presented [26]. The PCQ score significantly correlated with standardized health scores, indicating the severity of long COVID. In binary analyses, the PCQ score above the median was used as the cutoff.

Binary variables were constructed in accordance with previously published levels signifying either a positive screening result or a severe health outcome. The scoring system is furthermore described by Agergaard et al., 2022 [26].

We employed PCQ as the primary measure of the effect of vitamin D on the severity of long COVID, given vitamin D’s prior association with various health conditions [29,30].

Data were analyzed using Stata basic 17th edition. Prior to analysis, normal distribution of continuous variables was assessed using Q-Q plots. Frequency and means were reported, and results were compared as odds ratios (OR)s using logistic regression. Adjusted ORs were reported adjusting for age, sex, and smoking status, as these factors could potentially confound low vitamin D and severe long COVID. We applied Bonferroni correction to significance levels due to multiple comparisons. For Table 1, a corrected significance level of *p* < 0.003 was used, and for Table 2, a significance level of *p* < 0.002 was used. In our interaction analyses, we used the vitamin D levels measured at inclusion as a proxy for vitamin D levels at symptom onset.

## 3. Results

From 1 February 2021 to 1 November 2021, the outpatient Post-COVID Clinic at Aarhus University Hospital received and assessed 547 patients. Out of these, 447 patients gave their consent to participate in the study, and 442 underwent a vitamin D analysis. The mean age of the 447 participants was 47 years, with 28% (*n* = 127) being male, and 92% (*n* = 413) were of Danish ethnicity. Among the patients, 26% (115/442) had low vitamin D, with 100 having insufficient vitamin D and 15 having deficient vitamin D levels.

Table 1 provides a summary of the characteristics of the 447 participants, categorized into low and normal vitamin D groups.

Male participants had a higher likelihood of having low vitamin D compared with female participants (OR = 1.77, 95% confidence interval (CI) (1.12, 2.79), *p* = 0.014). The group with low vitamin D was younger, with a mean age of 41 years (standard deviation (SD) = 12), compared to the group with normal vitamin D levels, which had a mean age of 48 years (SD = 13) (OR = 0.96, 95% CI (0.94, 0.97), *p* < 0.001). When comparing the Charlson Comorbidity Index (CCI) of 0 versus ≥ 1 in the low vs. normal vitamin D groups, we found lower odds of comorbidity in patients with low vitamin D (OR = 0.47 (95% CI, 0.34–0.66)).

No significant differences were observed in the odds ratio between low vitamin D and normal vitamin D in the ethnically non-western European sub-cohort (OR = 1.24), 95% CI (0.47–3.3) compared to the ethnically Danish sub-cohort (OR = 0.74, 95% CI (0.35, 1.57)). No significant differences were found between vitamin D status and body mass index (BMI) when comparing patients with a BMI ≥ 25 vs. a BMI < 25. The numbers of participants consuming more than seven alcohol units per week and current smokers were low in both vitamin D groups, and the difference in vitamin D status was non-significant after applying the Bonferroni correction.

Table 2 presents long COVID symptoms in patients with low vitamin D and patients with normal vitamin D.

The prevalence of the most commonly occurring long COVID symptoms did not differ between the low and normal vitamin D groups (Table 2). In the low vitamin D group, the most frequently reported symptoms were disturbed sleep (89%, 94/106), physical fatigue (86%, *n* = 92/107), concentration difficulties (77%, *n* = 79/102), dyspnea at physical activity (65%, *n* = 70/107), and headaches (62%, *n* = 66/106).

In comparison, the normal vitamin D group reported disturbed sleep (83%, *n* = 261/314), concentration difficulties (83%, *n* = 257/308), physical fatigue (81%, *n* = 250/310), short-term memory problems (65%, *n* = 205/313), and headaches (62%, *n* = 196/314).

We observed some symptoms more commonly reported in the normal vitamin D group compared to the low vitamin D group, such as long-term memory problems (OR = 0.46, 95% CI (0.28, 0.77), *p* = 0.003), sore throat (OR = 0.39, 95% CI (0.17, 0.89), *p* = 0.026), nausea (OR = 0.47, 95% CI (0.23, 0.95), *p* = 0.035), and abdominal pain (OR = 0.44, 95% CI (0.20, 0.98), *p* = 0.044). These results were adjusted for age, sex, and smoking status. However, when applying the Bonferroni correction of significance at *p* < 0.002, no significant differences in reported symptoms between the two groups were found.

The odds of low vitamin D status were analyzed in various organ-specific symptom scores, based on a univariate logistic regression model. The CNS score (central nervous system score) is the sum of headache, dizziness, short-term memory problems, concentration difficulties, and paresthesia. The flavor score encompasses altered smell and taste. The CP score (cardiopulmonary score) consists of dyspnea at rest, dyspnea during exercise, cough, chest pain, and palpitations. The GI score (gastrointestinal score) is the sum of loss of appetite, altered stool habits, diarrhea, nausea, and abdominal pain. Lastly, the MS score (musculoskeletal score) is based on joint pain, joint swelling, myalgia, muscle exhaustion, and physical fatigue [26].

The odds of severely affected health in various outcome measures were compared between patients with low vitamin D vs. normal vitamin D status using a univariate logistic regression model. All variables were transformed into a binary format using the following cutoff values: PCFS (Post-COVID-19 Functional Status Scale) ≥ 3; OMC (Orientation–Memory–Concentration) < 25; extreme fatigue (Fatigue Assessment Scale) score ≥ 35; SCL depression score > 8; SCL anxiety score > 5; EQ-VAS score ≤ 50; PCQ (post-COVID symptom questionnaire) score > 35; MRC score (Medical Research Council dyspnea score) > 2; EQ index (EQ-5D-5L index) < 0.5 [26].

Figure 1 and Figure 2 depict various symptom scores and health scores, comparing patients with low vitamin D to those with normal vitamin D levels. No significant differences were observed in the symptom and health scores comparing vitamin D groups. As seen in Figure 2, the unadjusted OR for PCQ and vitamin D status was OR = 1.00, 95% CI (0.65–1.53). When adjusting for sex, age, and smoking status, we obtained an OR = 0.83, 95% CI (0.52–1.34).

We conducted an exploratory investigation to determine whether the results differed among sub-groups of patients with symptom onset during the summer season (April to September) and the winter season (January, February, March, October, November, and December). No significant differences in symptoms were found when comparing patients with low vitamin D to patients with normal vitamin D in these sub-groups. We also tested for effect modifications to see if seasonal changes interacted with the effect on low vitamin D status on our binary symptom and health scores. In these analyses, we found that our post-COVID-19 PCFS interacted with the season when patients experienced symptom onset (coefficient −0.597, *p* < 0.001). Patients with symptom onset in the winter season and low vitamin D had an increased likelihood of high PCFS, indicating a more severely affected functional status. This interaction was also observed with the depression variable (−0.446, *p* = 0.012). However, our PCQ did not show any significant interactions (Figure 3), and the same was true for the other health-related variables.

Figure 3 illustrates the distribution of PCQ scores stratified by the season of symptom onset and vitamin D status.

When assessing interactions between vitamin D and various health-related variables within different ethnic groups—non-western European immigrants, western European immigrant, and ethnic Danes—no significant difference was found.

Only 15 patients had vitamin D deficiency, and when comparing normal vitamin D levels with vitamin D deficiency, no significant differences in the health scores were observed.

## 4. Discussion

In our study population, we found that 26% of participants exhibited low vitamin D levels (25-OH D3 < 50 nmol/L). Those with low vitamin D levels tended to be younger, and low vitamin D levels were more prevalent among male participants. We did not find any association between low vitamin D and more severe long COVID. On the contrary, patients with normal vitamin D reported a higher occurrence of specific long COVID symptoms, such as long-term memory problems, nausea, and abdominal pain. However, when considering PCQ and standardized health scores, no significant differences were observed between the normal and low vitamin D groups. It is worth noting that patients with symptom onset during the winter season had a more severely affected functional level in cases of low vitamin D.

Hansen et al., 2018 [31] conducted a study in Denmark examining the seasonal variation of vitamin D in a Danish cohort, analyzing blood samples in both spring and autumn. They found that 86% of adults had sufficient vitamin D at some point during the year either in spring or autumn. However, only 49% maintained sufficient levels in both spring and autumn, highlighting the seasonal variation. In comparison, our study found that 74% of our study population had sufficient vitamin D values. Similar to our findings, Hansen et al., 2018, discovered that women and older participants had a higher median 25-OH D3 concentration than men and younger adults.

Comprehensive data reflecting the vitamin D status of the entire Danish population are not available. Nonetheless, based on various studies, it is estimated that around 86–96% of Danish adults have sufficient vitamin D levels during the summer season, whereas approximately 35–60% of Danish adults maintain sufficient vitamin D levels in the winter season [32,33,34]. Few Danish studies have examined vitamin D levels throughout the year in a healthy population, but with the large adult population (*n* = 2565) of Hansen et al., 2018 [31] a sufficient vitamin D level of 86% may be a reasonable estimate. This suggests that vitamin D insufficiency or deficiency was more frequent in our population (74%). Thus, an effect of low vitamin D on symptoms in our cohort might have been expected.

Our study, conducted in a Danish cohort, and a study by Mohamed Hussein et al. in 2022 [35] in an Egyptian long COVID cohort both reported a higher prevalence of low vitamin D in the study population compared to the respective prevalence of low vitamin D levels in each country. Mohamed Hussein et al. defined vitamin D values differently from us, where they considered 25-OH D3 values between 50 nmol/L and 75 nmol/L to be insufficient and 25-OH D3 < 50 nmol/L to be deficient. Only 4.6% (*n* = 10/219) of the participants in their study had normal vitamin D status, with 11% (*n* = 25/219) having insufficient and 84% (*n* = 184/219) deficient vitamin D status. In another study [36], 77% of healthy Egyptian women were found to be deficient in vitamin D, 14% insufficient, and 9% had normal levels, which is in contrast to the normal values of only 4.6% found by Mohamed Hussein et al. From these results, we cannot determine whether vitamin D levels play a role in acquiring long COVID, or if the lower vitamin D found in our populations is a result of the patients’ restrictions and constrictions associated with COVID-19 and long COVID, leading to a secondary reduction in vitamin D levels.

Mohamed Hussein et al., were also unable to establish an association between low vitamin D and long COVID symptoms. The symptom severity in this study was represented by a PCQ score, with a mean PCQ score of 15.9 ± 13.9 SD. The symptom score used in Mohammed Hussein et al. was previously described by Galal, I. et al. [37], which included 29 symptoms reported as absent, mild, moderate, or severe, ranging from a score of 0–87.

As shown in our baseline article [26], our participants’ health was severely affected, with functional disability and extreme fatigue being reported as moderate to severe by 33% and 62%, respectively.

In the study by Di Filippo et al. [38] (*n* = 100), lower vitamin D levels were observed at a six-month follow-up when comparing acute COVID-19 patients who developed long COVID against those who did not (50.17 vs. 57.9 nmol/L, *p* = 0.03). Additionally, lower vitamin D levels were observed among those who experienced neurocognitive symptoms.

When we examined the association with long-term memory problems, we observed that the group with low vitamin D had lower odds of experiencing long-term memory issues compared to the normal vitamin D group. The unadjusted odds ratio was OR = 0.53, 95% CI (0.33, 0.84), *p* = 0.007 and OR = 0.46, 95% CI (0.28, 0.77), *p* = 0.003 after adjusting for sex, age, and smoking status. This finding is unusual as it suggests that low vitamin D levels protect against long-term memory problems. In contrast, other observational studies have indicated that sufficient vitamin D levels are protective against age-related cognitive decline and the development of dementia [39,40]. Studies on vitamin D and cognition primarily focus on the geriatric population, whereas our cohort was younger, making direct comparisons challenging. 

Despite the existing literature suggesting a high prevalence of vitamin D insufficiency among individuals of non-western ethnic origins, our study did not observe such a trend. In a study by Andersen et al. [41], they found that 84% of women and 65% of men of Pakistani descent in Denmark had S-25OH levels below 25 nmol/L, and Andersen et al. [42] reported that 76% of refugees newly resettled in Denmark had vitamin D deficiency. A possible explanation for our contrasting findings may be a higher frequency of vitamin D supplementation among the non-western European participants in our study. Additionally, our sample size for non-western European participants was small, especially considering that more than 12% of the Danish population is of non-Danish ethnic origin, and the transmission of SARS-CoV-2 among ethnic minority groups was relatively high [43]. Participants referred to the Post-COVID Clinic may have had higher health literacy, better compliance with vitamin D supplementation, and better access to information on long COVID, which could have influenced them to seek referrals from their general practitioner.

### Limitations

Some important limitations of this study should be noted. Firstly, patients were selectively referred to the Post-COVID Clinic by their general practitioners based on the presence of complex and prolonged symptoms, resulting in a notably selected study population. Moreover, our limited sample size presents challenges, especially when drawing conclusions from sub-group analyses. Most of the reported symptoms were based on the patient’s subjective experiences, potentially introducing reporting bias, either through over-reporting or under-reporting of symptoms. However, there is little reason to believe that patients with low vs. normal vitamin D would report differently. Another significant limitation pertains to the lack of information on recent vitamin D supplementation, which might have obscured potential recent vitamin D insufficiency. It is plausible that participants experiencing chronic issues, such as long-term memory problems, might have been more likely to use vitamin D supplements.

Another significant limitation is the timing of vitamin D measurements. We examined the interaction of the season with symptom onset under the assumption that individuals with significantly low vitamin D levels would likely exhibit deficiency throughout the year. However, this is a notable constraint in these analyses, as vitamin D measurements were conducted at the time of presentation in the Post-COVID Clinic. Ideally, multiple blood samples should have been collected throughout the year to monitor vitamin D status, and information regarding vitamin D status at the time of infection would also have been preferable but was not available.

## 5. Conclusions

Our study found a higher prevalence of vitamin D insufficiency among long COVID patients compared with the background population. Despite the previous literature showing associations between vitamin D status and acute COVID-19 severity and mortality, we did not find similar associations between low vitamin D status and more severe long COVID symptoms in patients referred to a Post-COVID Clinic. To comprehensively understand the effect, if any, of vitamin D on long COVID, a randomized controlled trial designed to evaluate the effects of vitamin D supplementation on long COVID symptoms, with well-defined endpoints and a larger sample size, is warranted.

## Figures and Tables

**Figure 1 ijerph-20-07058-f001:**
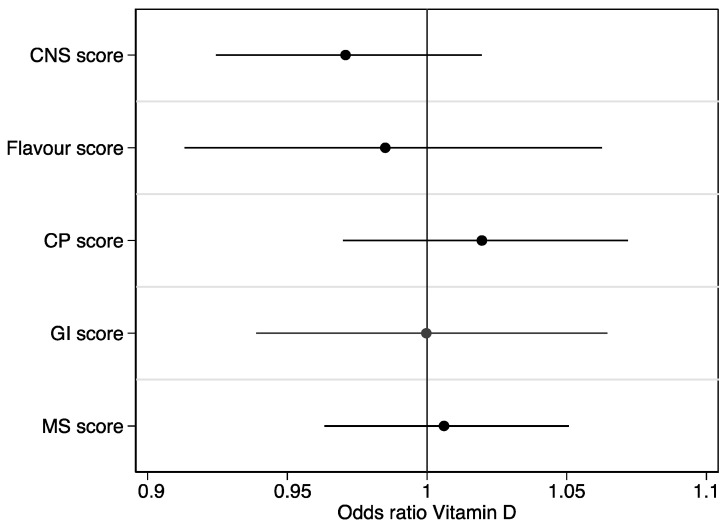
Organ-specific symptom scores compared in patients with low vitamin D to normal vitamin D levels. The odds of low vitamin D status analyzed in various organ specific symptom scores, based on a univariate logistic regression model. The CNS score (central nervous system score) is the sum of: headache, dizziness, short-term memory problems, concentration difficulties, and paresthesia. The flavor score encompasses altered smell and taste. The CP score (cardiopulmonary score) consists of: dyspnea at rest, dyspnea during exercise, cough, chest pain, and palpitations. The GI score (gastrointestinal score) is the sum of: loss of appetite, alternating stool habits, diarrhea, nausea, and abdominal pain. Lastly, the MS score (musculoskeletal score) is based on: joint pain, joint swelling, myalgia, muscle exhaustion, and physical fatigue [26].

**Figure 2 ijerph-20-07058-f002:**
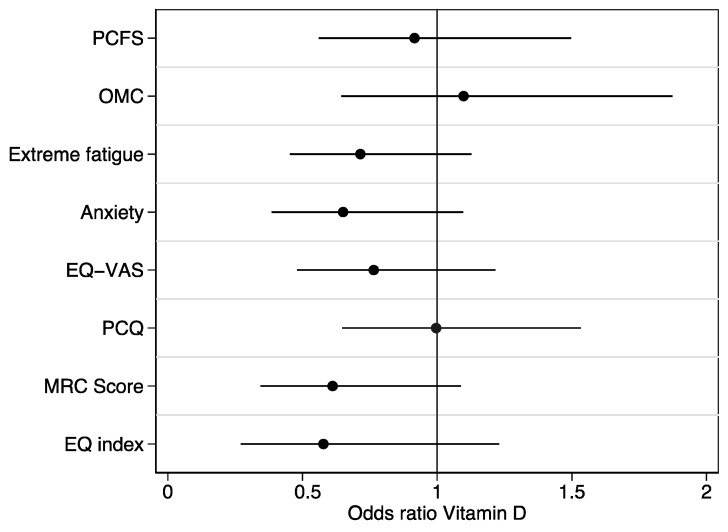
Health-related outcome scores compared in patients with low vitamin D and patients with normal vitamin D levels. Odds of severely affected health in various outcome measures compared between patients with low vitamin D vs normal vitamin D status using univariate logistic regression model. All variables were transformed into a binary format using the following cutoff values: PCFS (Post-COVID-19 Functional Status Scale) ≥ 3; OMC (Orientation-Memory-Concentration) < 25; Extreme fatigue (Fatigue Assessment Scale) score ≥ 35; SCL depression score > 8; SCL anxiety score > 5; EQ-VAS score ≤ 50; PCQ (post-COVID symptom questionnaire) score > 35; MRC score (Medical Research Council dyspnea score) > 2; EQ index (EQ-5D-5L-index) < 0.5 [26].

**Figure 3 ijerph-20-07058-f003:**
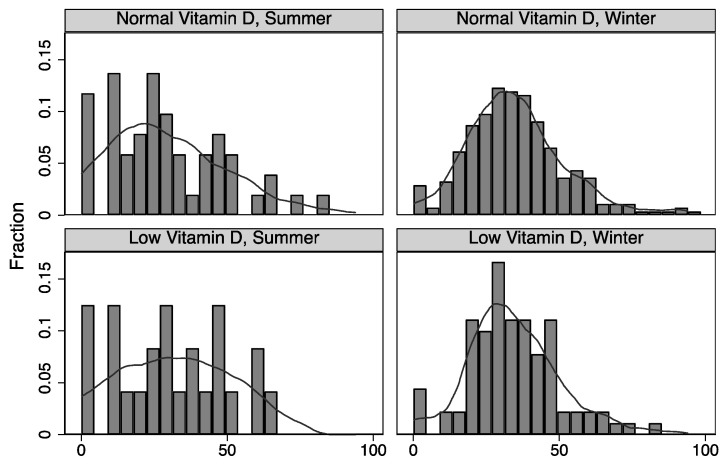
Seasonal effect modification.

**Table 1 ijerph-20-07058-t001:** Demographic and previous clinical characteristics of long COVID patients in a Post-COVID Clinic—univariate analysis comparing patients with low vitamin D vs. normal vitamin D.

	Low Vitamin D 26% *n* = 115/442	Normal Vitamin D 74% *n* = 327/442	Unadjusted OR Low Vit D vs. Normal Vit D (95% CI)
**Sex (male)**	37% (42/114)	25% (81/327)	**1.77 (1.12–2.79)**
**Age, mean (SD)**	41 (12) years	48 (13) years	**0.96 (0.94–0.97)** *
**Other ethnic origin than Danish**	10% (11/114)	7% (24/327)	1.35 (0.64–2.85)
**Non-western European ethnic origin**	5% (6/114)	4% (14/327)	1.24 (0.47–3.31)
**Familiar dispositions**	34% (39/114)	44% (145/327)	0.65 (0.42–1.02)
**Autoimmunity**	8% (9/114)	13% (44/327)	0.55 (0.26–1.17)
**Metabolic**	21% (24/114)	25% (81/327)	0.81 (0.48–1.36)
**Allergy**	13% (15/114)	18% (58/327)	0.70 (0.38–1.30)
**Transmission of SARS-CoV-2**			
**Work**	40% (46/114)	37% (122/327)	1.14 (0.73–1.76)
**Family**	29% (33/114)	25% (83/327)	1.20 (0.74–1.93)
**Travel**	<1%	4% (14/327)	0.40 (0.09–1.78)
**Unknown**	18% (21/114)	18% (60/327)	1.00 (0.58–1.74)
**Other**	14% (16/114)	14% (47/327)	0.97 (0.53–1.80)
**Time from symptom onset to baseline, mean (SD), months**	6.9 (3.4)	7.7 (3.8)	1.0 (0.996–1.00)
**SARS-CoV-2 infection (winter)**	77% (84/109)	81% (227/282)	0.81 (0.48–1.39)
**No positive PCR or AB test**	<1%	3% (10/327)	0.28 (0.04–2.22)
**Hospitalized ^1^**	11% (13/114)	12% (40/327)	0.92 (0.47–1.80)
**Oxygen supplementation**	63% (5/8)	56% (10/18)	1.33 (0.24–7.35)
**Charlson comorbidity index**			**0.47 (0.34–0.66)** *
**0**	77% (88/114)	50% (163/327)	
**≥1**	23% (26/114)	50% (164/327)	
**Diabetes**	<1%	2% (5/327)	1.74 (0.41–7.40)
**Asthma**	14% (16/114)	15% (48/327)	0.95 (0.52–1.75)
**COPD**	<1%	2% (7/327)	0.40 (0.05–3.32)
**CHD**	<1%	<1%	1.93 (0.32–11.69)
**CVD**	<1%	<1%	1
**Hypertension**	12% (14/114)	10% (34/327)	1.21 (0.62–2.34)
**Malignancy (previous)**	<1%	4% (14/327)	0.20 (0.02–1.52)
**Connective tissue disease**	<1%	2% (5/327)	1
**Immunodeficiency**	<1%	<1%	2.88 (0.18–46.5)
**Previous depression**	9% (10/114)	14% (47/327)	0.57 (0.28–1.18)
**Previous stressful episode**	11% (12/114)	13% (43/327)	0.78 (0.40–1.53)
**Medicine**			
**ACE or AT2 rec inhibitor**	12% (13/111)	11% (35/318)	1.07 (0.55–2.11)
**Statins**	8% (9/111)	11% (36/317)	0.69 (0.32–1.48)
**Steroids**	<1%	2% (5/316)	1
**Other immunosuppressant**	<1%	3% (10/316)	1
**Current smoker**	12% (13/107)	6% (17/308)	**2.37 (1.11–5.06)**
**Alcohol, >7 per week**	9% (10/107)	4% (12/301)	**2.48 (1.04–5.93)**
**BMI ≥ 25**	66% (73/111)	63% (200/316)	1.11 (0.71–1.75)
**Socioeconomic status**			
**Education**			
**Primary school**	12% (14/114)	8% (26/327)	1.62 (0.81–3.22)
**High school**	16% (18/114)	15% (48/327)	1.09 (0.60–1.96)
**Vocational education**	31% (35/114)	22% (73/327)	1.54 (0.96–2.48)
**Medium-term higher education**	32% (36/114)	44% (143/327)	**0.59 (0.38–0.93)**
**Long-term higher education**	11% (12/114)	9% (31/327)	1.12 (0.56–2.27)
**Master/PhD**	<1%	3% (11/327)	0.51 (0.11–2.35)
**Work**			
**Employed**	82% (93/114)	81% (264/327)	1.06 (0.61–1.83)
**Self employed**	4% (5/114)	6% (19/327)	0.74 (0.27–2.04)
**Student**	9% (10/114)	4% (12/327)	**2.52 (1.06–6.01)**
**Retired**	<1%	7% (22/327)	0.50 (0.17–1.50)
**Not living alone**	90% (84/93)	80% (220/275)	1.17 (0.99–1.40)
**Size of housing, mean (SD) m^2^**	146 (67)	142 (62)	1.00 (0.997–1.005)

Abbreviations: COPD = chronic obstructive pulmonary disease, CHD = coronary heart disease, CVD = cerebrovascular disease, BMI = body mass index. ^1^ Acute COVID-19 hospitalization values below five were omitted due to ethical reasons. Bold text indicates *p* < 0.05. An asterisk (*) symbol indicates Bonferroni correction for the significance level: *p* < 0.003.

**Table 2 ijerph-20-07058-t002:** Patient-reported symptoms at evaluation in a Post-COVID Clinic—univariate analysis comparing patients with low vitamin D vs. normal vitamin D.

Long COVID Symptoms	Some to Very Much		Odds Ratio (OR)	
During 4 Weeks Previous to Evaluation in a Post-COVID Clinic	Low Vitamin D *n* = 115/442	Normal Vitamin D *n* = 327/442	Unadjusted OR Low Vit D vs. Normal Vit D	Adjusted ^1^ OR Low Vit D vs. Normal Vit D
**Headaches**	62% (66/106)	62% (196/314)	0.99 (0.63–1.56)	0.86 (0.51–1.43)
**Dizziness**	38% (40/106)	37% (116/313)	1.03 (0.65–1.62)	0.89 (0.54–1.47)
**Paresthesia**	27% (29/106)	27% (82/308)	1.04 (0.63–1.70)	0.94 (0.55–1.60)
**Concentration difficulties**	77% (79/102)	83% (257/308)	0.68 (0.39–1.19)	0.60 (0.33–1.09)
**Short-term memory problems**	59% (63/106)	65% (205/313)	0.77 (0.49–1.21)	0.68 (0.42–1.11)
**Long-term memory problems**	32% (34/107)	46% (145/313)	**0.53 (0.33–0.84)**	**0.46 (0.28–0.77)**
**Impaired smell**	36% (38/106)	38% (118/314)	0.93 (0.59–1.47)	0.98 (0.60–1.59)
**Impaired taste**	35% (37/106)	34% (106/312)	1.04 (0.66–1.66)	1.05 (0.64–1.73)
**Runny nose or nasal congestion**	23% (24/104)	23% (71/311)	1.01 (0.60–1.72)	0.81 (0.45–1.44)
**Sore throat**	10% (10/105)	16% (50/308)	0.54 (0.26–1.11)	**0.39 (0.17–0.89)**
**Cough**	20% (21/105)	16% (50/307)	1.29 (0.73–2.26)	1.14 (0.61–2.11)
**Expectoration**	14% (15/106)	11% (34/311)	1.34 (0.70–2.58)	0.98 (0.47–2.05)
**Dyspnea at rest**	31% (33/106)	29% (90/314)	1.13 (0.70–1.82)	1.01 (0.60–1.69)
**Dyspnea at physical activity**	65% (70/107)	60% (189/315)	1.26 (0.80–1.99)	1.06 (0.65–1.74)
**Chest pain**	32% (34/107)	26% (82/312)	1.31 (0.81–2.11)	1.02 (0.61–1.72)
**Palpitations**	36% (39/107)	28% (86/312)	1.51 (0.95–2.40)	1.17 (0.70–1.95)
**Loss of appetite**	23% (23/98)	16% (47/288)	1.57 (0.90–2.76)	1.09 (0.58–3.8)
**Nausea**	12% (13/107)	19% (58/311)	0.60 (0.32–1.15)	**0.47 (0.23–0.95)**
**Diarrhea**	12% (13/105)	9% (27/308)	1.47 (0.73–2.97)	1.10 (0.51–2.37)
**Abdominal pain**	10% (10/105)	15% (45/309)	0.62 (0.30–1.27)	**0.44 (0.20–0.98)**
**Altered bowel habits**	26% (27/102)	28% (86/306)	0.92 (0.56–1.53)	0.85 (0.49–1.46)
**Skin rash**	11% (11/104)	7% (22/301)	1.50 (0.70–3.21)	1.39 (0.62–3.13)
**Itching skin**	14% (15/105)	15% (47/307)	0.92 (0.49–1.73)	0.85 (0.43–1.67)
**Joint pain**	38% (41/107)	41% (126/304)	0.88 (0.56–1.40)	0.99 (0.61–1.61)
**Swollen joints**	17% (17/103)	7% (22/301)	**2.51 (1.27–4.94)**	**2.86 (1.36–6.01)**
**Myalgia**	48% (50/105)	49% (150/308)	0.96 (0.61–1.49)	1.05 (0.65–1.69)
**Muscle exhaustion**	58% (58/100)	54% (164/302)	1.17 (0.74–1.85)	1.14 (0.70–1.86)
**Physical fatigue**	86% (92/107)	81% (250/310)	1.47 (0.80–2.72)	1.39 (0.72–2.69)
**Subjective fever**	8% (8/95)	14% (39/276)	0.56 (0.25–1.24)	**0.35 (0.14–0.89)**
**Disturbed sleep**	89% (94/106)	83% (261/314)	1.59 (0.81–3.11)	1.39 (0.69–2.80)
**Problems falling asleep**	37% (40/107)	42% (132/315)	0.83 (0.53–1.30)	0.66 (0.40–1.08)
**Awakening**	53% (56/106)	58% (179/311)	0.84 (0.54–1.31)	0.85 (0.53–1.39)

Symptoms were considered present when the patient reported some, a lot or very much. ^1^ Age, sex, and smoking status. Bold text indicates *p* < 0.05.

## Data Availability

Data were registered in a secured REDCap database at https://redcap.au.dk/ accessed on 14 March 2022. The data are not publicly available due to ethical restrictions.

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
