# Peer review of "Effect of Vitamin D in Long COVID Patients"

_ijerph, 2023, doi:10.3390/ijerph20227058_

Round 1
Reviewer 1 Report
Comments and Suggestions for Authors
This is an interesting evaluation of vitamin D status in patients with long COVID symptoms. It is important that research with so-called “negative” findings is published.
The paper is clearly written. The methods and results are possibly easier to understand fully if one has already read the earlier publication from the same group. The results are fairly dense.
Comments are not major and generally point our minor issues to enhance clarity.
I defer to other reviewers regarding the veracity of the statistical method and reporting thereof.
Abstract: “of which 547 patients were referred “ – needs to specify where they were referred to.. e.g., “referred to a specialized long COVID-19 clinic”.
Table 1 & 2 titles should possibly state whether these variables were analyzed as univariate (presumably that is the case).
L138 “and admitted” sounds as if they were all hospitalised, rather than being “outpatients”. I presume the concept is that they were accepted for assessment by the clinic.
L142-3 “Conversely, 74% (327/442) of the patients had normal vitamin D values.” can be deleted, as it is just the corollary of L141.
L:285-6 it is not clear what the “study population” is in the Egyptian study – presumably they were also long covid sufferers?
L324 Charlson Comorbidity Index – may need further explanation; its only other mention is in the Table. The focus on this finding may not be needed in the text, given that “this was not significant when adjusting for age, sex, and smoking status.”
L 349 Ideally, multiple blood samples should have been collected throughout the year to 349 monitor vitamin D status – this could be listed under Limitations.
Comments on the Quality of English Language
The language quality is high. However, a number of minor improvements should be made. For example:
“Vitamin” is sometimes capitalized & sometimes not – needs to be consistent (probably uncapitalized).
Fig 1 & 2 titles each have two spelling errors.
L108 & 109: minor English editing needed
L 114-117 need further punctuation inserted.
L123 minor English editing needed
L235-6-7 minor English editing/punctuation needed
L366 minor English editing needed.
Reviewer 2 Report
Comments and Suggestions for Authors
It is a well-written article and a topic of contemporary importance. I appreciate that the authors were candid on the limitations of this hospital-based study involving minimal subjects. The authors could study only 442 out of 547 (less than 80%) of the post-COVID-19 patients admitted in that sentinel hospital during the study period of ten months (Feb-Nov 2021). Still, many segregated analyses with Odds ratio estimations are being attempted(Table 1). Some numbers in the subgroups are insufficient to draw any meaningful conclusions. Then again, trying a seasonal trend analysis, where they do not even have a full year of data, is unacceptable in scientific terms. So, one of the study's conclusions of not finding any difference in vitamin D insufficiency with seasons, which contradicts the previous studies, is not valid in scientific terms.
I suggest that the authors be modest in acknowledging the limited number of study subjects and the possibility of selection bias in this hospital-based study before contradicting the well-established facts.
Reviewer 3 Report
Comments and Suggestions for Authors
Dear Editor,
I carefully read the manuscript by Agergaard et al.
My comments and suggestions for the authors are the following:
- All the abbreviations should be defined at their first occurrence in the manuscript (and in the abstract too).
- English language needs to be carefully revised and improved. In general, the style should be made more fluid, because now the manuscript is quite difficult to read.
- Line 89: The title should be revised in "Data collection". In effect, information regarding "Data analysis" has been included in the in the paragraph 2.3.
- Lines 130-134: The authors should include more details as regards the statistical analysis. For example, they should specify how the normal distribution of the variables was assessed. Levels of significance (p-value) for the analysis should also be specified here.
- Table 1: "Coronary heart dis." should be more properly abbreviated as "CHD".
- Table 1: "Cerebrovascular dis." should be more properly abbreviated as "CVD".
- Tables: All the abbreviations included in the table should be specified at the bottom of the table.
- Line 368: "a clinical trial is warranted". The authors should be more precise here.
- The authors should highly consider to refer to doi: 10.1016/j.clnu.2023.09.008, doi: 10.3390/nu14173584 and doi: 10.3389/fimmu.2023.1231813 in their manuscript.
- The limitations of the study should be further and more deeply discussed.
Comments on the Quality of English LanguagePlease, see my comments below.
Round 2
Reviewer 3 Report
Comments and Suggestions for Authors
Dear Editor,
I carefully read the revised version of the manuscript that is significantly improved compared to the original version.
Author Response
Thank you for acknowledging the improvements in the revised manuscript